# Causal inference and cognitive-behavioral integration deficits drive stable variation in human punishment sensitivity

Lilith Zeng [1], Haeme R. P. Park[1,2], Gavan P. McNally [1] & Philip Jean-Richard-dit-Bressel [1] ✉

Some individuals persist in behaviors that incur harm to themselves or others. While adaptive decision-making requires integrating such punishment feedback to update action selection, the mechanisms driving individual differences in this capacity remain unclear. Here, in a sample spanning 24 countries (*N* = 267), we used a conditioned punishment task to identify how individuals learn from and adapt to punishment. We identified three, behaviorally robust phenotypes: (1) *Sensitive*, who correctly inferred punishment causality and adaptively updated decisions through direct experience of punishment; (2) *Unaware*, who failed to correctly infer punishment causality from direct experience but corrected their decisions following an informational intervention clarifying consequences; and (3) *Compulsive*, who persisted in harmful decisions despite both punishment and informational intervention. These phenotypes were driven by distinct cognitive mechanisms: (1) *causal inference deficits*, where individuals misinterpreted punishment causality, impairing correct knowledge acquisition (remediable via targeted informational intervention); and (2) *integration failure*, a deficit in synthesizing causal knowledge, action valuation, and action selection that rendered decision-making inert to punishment feedback, even after targeted informational intervention. Remarkably, these phenotypes predicted longitudinal outcomes (learning trajectories, choice behavior) six months later. By identifying the cognitive mechanisms driving variation in human punishment learning, this work provides a framework to understand why individuals persist in harmful behavior and highlights the need for approaches to address these distinct cognitive barriers to adaptive decision-making.

A core tenet of adaptive behavior is that rewards reinforce actions, while punishments suppress them. This tenet underpins shaping of individual choices, artificial intelligence and machine learning, as well as societal systems designed to promote co-operative behavior[1–3]. Yet, this tenet fails to explain why many individuals persist in punished behaviors, incurring profound costs from their choices[4,5]. Such punishment insensitivity can manifest early in life[6], predict adverse outcomes across health and well-being[8–9], and resist interventions ranging from fines to incarceration[10,11]. Why do some individuals adaptively avoid harm through experience, while others exhibit maladaptive behavioral persistence?

Current explanations for this variation in punishment sensitivity remain fragmented. Studies of punishment sensitivity have focused on learning in clinical populations, incarcerated individuals, individuals with addiction, brain lesions, or neurodegeneration[10–15], leaving the spectrum of individual differences in the wider population poorly

mapped. Compounding this, fundamental questions about punishment sensitivity persist. The mechanisms that drive these differences between individuals remain poorly understood, so it remains unclear whether punishment-insensitive individuals fail to learn from consequences, misvalue consequences, or struggle to act on their acquired knowledge about punishment[16–19]. It is also unclear whether observed variations in punishment sensitivity reflect stable phenotypic differences in learning and decision-making or random fluctuations in behavior[20]. Mechanistic answers to these questions are essential to improving decision-making in contexts ranging from public health campaigns to cognitive-behavioral therapies.

Here, we address these gaps through three advances. First, we assess punishment learning in a general population sample spanning 24 countries. Second, we employ a task[4,5,21,22] that precisely isolates what individuals learn from punishment. Third, we test whether differences between individuals

¹School of Psychology, UNSW, Sydney, NSW, Australia. ²Neuroscience Research Australia, Sydney, NSW, Australia.
✉e-mail: p.jean-richarddintbressel@unsw.edu.au

predict learning and decision-making over a six-month interval. We identify robust phenotypes that reflect stable, trait-like differences in causal inference and cognitive-affective-behavioral integration, highlighting the need for approaches to address the cognitive barriers to adaptive decision-making.

## Methods

The study was approved by UNSW Human Research Ethics Advisory Panel C (HREAP-C #3385). Informed consent was obtained from all participants. This study was not pre-registered.

### Design

Data collection took place in two stages: an initial test and a retest set six months later. During each stage, participants filled out self-report questionnaires, and played six 3 min blocks of a Planets & Pirates game[5,22] interspersed with post-block self-report measures (see *Post-block measures*) and task information (see *Procedures*).

**Game design.** The game is based on the Fixed Utility protocol[22]. Two continuously-displayed planets (R1, R2) that could be "traded" with (clicked on) for point rewards. In initial pre-punishment blocks, R1 and R2 clicks yielded rewards (+100 points) with 50% probability (2 sec countdown). Rewards for R1 and R2 responses were independent; both planets could be in countdown simultaneously so maximising point gain during pre-punishment involved clicking on both planets.

In later punishment blocks, responses continued to yield rewards but could also trigger ship cues (CS+ , CS–). R1 exclusively triggered CS+ while R2 exclusively triggered CS–. CS+ led to an "Attack" (substantial point loss), whereas CS– had no negative consequence. Despite the ongoing reward contingency, the R1 → CS+ →Attack contingency meant R1 responses generally had negative utility during punishment blocks, while the safe R2 → CS– contingency meant R2 responses maintained its positive utility. Therefore, maximising point gain in punishment blocks involved avoiding R1 responses in favor of R2 responses. A shield button was present during ship presentations, but was only available to be selected on 50% of ship trials. If selected (-50 point cost), the shield prevented an upcoming attack. This mitigated, but did not nullify the overall negative utility of R1 responses. Given the relative rarity of shield events (many participants did not receive the opportunity to use a shield for a given CS within a block), shield measures were not examined.

Following 3 blocks of punishment, R1 → CS+ →Attack and R2 → CS– contingencies were revealed to participants (see *Apparatus and Stimuli*), and they were given a final punishment block to assess effects of this reveal on behavior and cognitions.

Participants were randomly allocated to 10% or 40% Action-CS probability groups at the start of the initial test. Planet clicks triggered CSs with 10% or 40% probability for 10% and 40% groups, respectively. To match the negative utility of punished R1 responses across probability groups, attacks caused –40% and –10% point loss (of accumulated points) for 10% and 40% groups, respectively.

**Retest.** Individuals that passed initial test engagement checks (see *Participants*) were invited undertake the retest. All participants were allocated to their original probability group, but other counterbalancing parameters were randomised (e.g., punished R1 = left vs. right planet, CS+ = ship Type 1 vs. 2). All other aspects of retest (design, procedure) were identical to the initial test.

### Participants

Participants were recruited from the online research platform Prolific[23,24]. Given that understanding task instructions was essential for participation, self-reported fluency in English was a selection criterion. Participants were reimbursed pro-rata with monetary compensation for their time (£9/hour). Participants were informed that Top 10% scorers would be receive a monetary bonus. Bonuses (additional £10) were determined and distributed after each stage of data collection (initial test and retest).

Participants were excluded from analyses if they failed either of two engagement checks: (1) failing to give correct responses to two catch questions embedded in the questionnaire battery (see *Self-reported trait questionnaires* for details); or (2) answering post-block measures too quickly or slowly (<0.8 s or >30 s per question, averaged per page). These were included to minimize the issue of inattentive participants driving reported effects[25]. For the initial test, 267 participants (aged 18-63 years; 118 female, 143 male, 6 other) met inclusion criteria. For retest, 128 participants (aged 20–60 years; 59 female, 67 male, 2 other) met inclusion criteria.

### Apparatus and stimuli

The experiment was programmed using the jsPsych library[26]. Experiment code can be found at https://github.com/philjrdb/HCP-Test-Retest and https://zenodo.org/records/155819599.

**Game interface.** During game blocks, participants had control of a custom mouse pointer that turned dark when clicking (visual feedback). Two planets were continuously displayed center-left and center-right of the screen. The color (orange vs. blue) and location (left vs. right) of punished R1 vs. unpunished R2 planets was counterbalanced across participants. A green ring appeared around a planet whenever the pointer hovered over it (visual feedback). Clicks on either planet (R1, R2) triggered a trading signal cue (2 s), displayed directly beneath each planet. Trading signal terminated with reward ("Success! +100" in green text) or no reward ("Trade attempt failed" in yellow text) outcome, displayed directly above each planet. Accumulated points were continuously displayed top-center of the screen.

"Incoming ship" icons (Type I [turquoise], Type II [purple]) were presented in the upper-middle part of the screen. A countdown timer (6secs) was presented immediately below the ship icon. After 6secs, ship outcomes (attack ["Attack!" with point loss amount in red text]; nothing ["Ship passed by without incident" in green text]) were presented for 2 s below the ship icon. A shield indicator/button was displayed in the lower-middle part of the screen during ship presentations; this button indicated if a shield was available or unavailable during a given ship trial. If available, the button could be clicked to "activate" the shield ("-50 points" cost shown above button). If activated during a CS+ trial, the subsequent attack was prevented ("Attack deflected" in gray text shown in place of attack outcome), whereas activating a shield had no effect on CS– outcomes.

**Reveal.** Prior to the last punishment block, task contingencies were revealed to participants. This involved a page describing and depicting R1 → CS+ → Attack and R2 → CS– as follows:

"Local intel has determined where the pirates are coming from!

Your signals to the [blue/orange] planet ([left/right] side) have been attracting pirate ships (Ship: [Type 1/Type 2]), that have been stealing your points!

Your signals to the [orange/blue] planet ([right/left] side) have only been attracting friendly ships (Ship: [Type 2/Type 1])."

A depiction of each contingency (R1 → CS+ → Attack; R2 → CS– [using relevant planet/ship/attack icons and arrows]) was displayed beneath each contingency statement.

### Post-block measures

Valuations, inferences, preference estimates, and preference endorsements were polled via labeled 100-point sliders after each game block. The default slider position was set at 50 (midpoint).

**Valuations.** Participants were asked how they felt about game elements ("Very negative" to "Very positive"). Each game element (planets, ships, outcomes) was indicated with an icon and label.

**Causal inferences.** Participants were asked how often interacting with a game element led to another game element ("Never [0%]" to "Every time

[100%]"). Inferences per antecedent (R1, R2, Ship I, Ship II) were assayed on separate pages. The antecedent icon was displayed at the top of the screen, and each slider for each potential consequence (e.g., ships, outcomes) were accompanied by an icon and label.

**Preference estimates.** Participants were asked to reflect on the most recent block and estimate the proportion of their clicking across planets ("100% Planet A/0% Planet B" to "0% Planet A/100% Planet B"). Relevant planet icons were displayed at the two ends of the slider.

**Preference endorsements.** Participants were asked to reflect on the optimal strategy for maximizing points in the preceding block ("100% Planet A/0% Planet B" to "0% Planet A/100% Planet B"). Relevant planet icons were displayed at the two ends of the slider.

### Self-reported trait questionnaires
Participants completed a self-report questionnaire battery consisting of Cognitive Flexibility Index (CFI)[27], Habitual Tendency Questionnaire (HTQ)[28], and Alcohol Use Disorders Identification Test (AUDIT)[29]. Each questionnaire was presented on separate pages. Catch questions were embedded halfway through the CFI ("Select the left-most option, strongly disagree, for this question") and AUDIT ("Select the fourth option from the left, weekly, for this question").

**CFI.** The CFI is a measure of one's ability to think adaptively when encountering difficult situations[27]. The questionnaire consists of 20 items rated on a seven-point scale ranging from 1 ("strongly disagree") to 7 ("strongly agree"). Items are divided into two subscales: "Control", which measures self-reported tendencies to view challenging situations as relatively controllable, and "Alternatives", which measures self-reported ability to perceive and generate alternative solutions to a problem. Higher scores indicate greater cognitive flexibility.

**HTQ.** The HTQ assesses inclinations to enact repeated behavior without conscious thought or effort[28]. The questionnaire contains 11 items rated on a seven-point scale ranging from 1 ("strongly disagree") to 7 ("strongly agree"). Items are divided into three subscales: "Compulsivity", preference for "Regularity", and "Aversion to novelty". Higher scores indicate greater habitual tendencies.

**AUDIT.** The AUDIT is a measure of alcohol consumption patterns over the past 12 months. It consists of 10 items rated on a five-point scale ranging from 0 to 4. Higher scores indicate greater alcohol consumption and risk of problematic drinking.

### Procedure
**Informed consent and self-reported demographics.** Participants were provided with an information sheet with consent form. After informed consent was obtained, participants were asked to report their gender ("Male", "Female", "Other"), age, and native language (not used as selection criteria). No data on race/ethnicity was collected.

**Self-report questionnaires.** Participants received the self-report questionnaire battery. Participants that correctly answered catch questions continued with the experiment, while those that did not were ejected and prevented from re-participating.

**Initial instructions.** Participants were then informed that they would play a game over 6 blocks and that the goal of the game was to gain as many points as possible. They were informed they could gain points by "trading" with planets by clicking on them. Participants were also informed that top 10% scoring participants would earn an additional monetary prize. Participants could not proceed with the experiment until they passed a brief multiple-choice comprehension test.

**Pre-punishment phase.** Participants underwent two 3 min game blocks, each followed by relevant post-block measures. During game blocks, R1 and R2 responses could earn point rewards (each +100, 50% probability).

**Punishment (pre-reveal) phase.** Participants were informed of local pirates that might steal points from them. They were also informed that a shield button could be clicked to prevent attacks, but that it was not always available. Participants were reminded that their goal was to maximise points before proceeding with the experiment. Participants then underwent three 3 min game blocks, each followed relevant post-block measures. During game blocks, R1 responses could yield reward as well as CS+ ships, and R2 responses could yield reward as well as CS– ships. CS+ co-terminated with attack (point loss), whereas CS– had no negative consequence.

**Reveal.** Participants were informed of R1 → CS+ → Attack and R2 → CS−contingencies (see *Apparatus and Stimuli*). Encoding of this information was confirmed with a follow-up multiple-choice contingency knowledge test; participants were returned to the previous contingency reveal page until they answered the contingency knowledge test correctly.

**Post-reveal phase.** After the reveal, participants received a final punishment block with post-reveal measures (identical to pre-reveal punishment).

### Post-retest reflections (retest only)
To assess awareness of test-retest performance, participants were asked 3 questions (displayed on separate pages) at the end of the retest experiment: Responses were assessed via 100-point sliders [text displayed beneath 0 and 100, respectively]:
1. "While playing the game, did you remember playing the same game in the past?"
   ["I don't think I have played it before"; "I remembered everything about this game"]
2. "Based on this memory, did you choose to change how you played the game this time?"
   ["I did not change anything"; "I completely changed my strategy."]
3. "Do you think you did better or worse playing the game **this time** compared to your performance last time?"
   ["I did much worse this time"; "I did much better this time."]

### Data analysis
De-identified experiment data is provided at https://osf.io/ju35h/. Data was extracted and pre-processed using custom MATLAB scripts (https://github.com/philjrdb/HCP-Test-Retest, https://zenodo.org/records/155819599). Clustering, chi-square, ANOVAs, t-tests, and stepwise logistic regression were performed in SPSS (version 29.0). Confidence intervals for ANOVA effect sizes (partial eta squared [$\eta_p^2$]) were calculated in R using *apaTables* package (version 2.0.8). Where applicable, evidence for the null (BF$_{01}$) was assessed using Bayesian ANOVA and *t* tests in jamovi (v.2.6.44.0; jsq module v.1.2.0). Singular Value Decomposition was performed in MATLAB (R2022b). Given no overall differences in pre-punishment data (Fig. S2), all data for pre-punishment blocks were collapsed, such that "Pre" represents averaged pre-punishment phase data.

**Task behavior.** Participant behavior was assessed via R1 and R2 click rates (clicks/min) during non-CS periods. These rates were used to calculate a self-normalized preference score [(R1 rate/Overall rate)*100] indicating the percentage of clicks that were R1. A score of 50% indicates equal rates of R1 and R2 (i.e., no preference) whereas score of 0 indicates a complete preference for R2 over R1. Differences in behavior (click rates, preferences) were analyzed using orthogonal contrasts (see *Contrast analysis* below). Significant bias in preference was also determined via one-sample t-tests against the null value of 50.

**Clustering.** Behavioral phenotypes were auto-identified using TwoStep clustering (Bayesian Information Criterion; Table S1), using final pre-reveal and post-reveal preferences as inputs. Clustering was performed separately for initial test and retest to allow data-driven discovery of phenotypes per test; the same 3 phenotypes (Sensitive, Unaware, Compulsive) were found per test. Phenotype stability (test-retest cluster change) was examined by categorizing individuals as having the same avoidance phenotype (Sensitive→Sensitive, Unaware→Unaware, Compulsive→Compulsive), an improved avoidance phenotype (Unaware→Sensitive, Compulsive→Sensitive, Compulsive→Unaware), or a worse avoidance phenotype (Unaware→Compulsive, Sensitive→Unaware, Sensitive→Compulsive) at retest.

**Contrast analysis.** Behavior and post-block measure data were analyzed via between- x within-subject ANOVAs using orthogonal contrasts. Where applicable, within-subject contrasts were block (linear), action (R1 vs. R2), CS (CS+ vs. CS −), and/or inferences (correct vs incorrect R → CS). Block contrasts used all valid blocks ([Pre,Pun1,Pun2,Pun3,Rev]) unless indicated as "Pre-reveal" (applicable [Pre,Pun1,Pun2,Pun3]) or "Reveal" ([Pun3,Rev]). Between-subject factors were cluster (Sensitive vs. Unaware vs. Compulsive), probability group (10% vs. 40%), and/or test-retest cluster change (Same vs. Improve vs. Worse). Where applicable, follow-up analyses were conducted using one-way ANOVAs, pairwise between-subject comparisons (Sidak correction), and/or t-tests of a within-subject factor (e.g. R1 vs. R2) per cluster. In the case of key non-significant results, evidence in favor of the null ($BF_{01}$) was assessed using Bayesian ANOVA and t-tests. Confidence intervals for $\eta_p^2$ was calculated at the recommended 90% level[30].

**Chi-square.** Significant interactions between categorical variables (cluster, probability group, sex, retested vs. not) were assessed via Pearson Chi-square (2-sided).

**Cognitive-behavioral trajectories.** Singular value decomposition (SVD) was used to represent relationships between action-related bias measures (R1:R2 attack inferences, valuation, endorsement, estimate, behavior) per cluster. This approach does not rely on assumptions of normality, homoscedasticity, or directionality. Rather, it holistically captures functional "true-score" relationships between variables within a single analysis.

R1:R2 measures per block per individual in a cluster was used as input for SVD analyses. Given all R1:R2 measures except attack inferences were measured during pre-punishment blocks, the null value of 50 (no bias) was imputed for pre-punishment R1:R2 attack inferences so pre-punishment data could contribute to the analysis. Data per cluster was organized into matrices (rows = individuals' block data; columns = R1:R2 measures) and centered by subtracting column means. The centered matrix ($A$) was then decomposed into three matrices ($U$, $\Sigma$, and $V^T$) using MATLAB *svd* function, such that $A = U\Sigma V^T$. $V^T$ contains the principal axes of data covariance (eigenvectors). The prime eigenvector in $V^T$ captured the majority of covariance across R1:R2 measures per cluster (Sensitive $r^2$: 74.0%; Unaware $r^2$: 73.5%; Sensitive $r^2$: 57.3% [calculated via MATLAB *pca* function]), indicating a single linear trajectory captured the majority of cross-measure variance.

The reliability of prime eigenvectors and cluster differences in these was determined by bootstrapping cluster data (resampling subjects with replacement) to obtain 1000 bootstrapped $V^T$ per cluster. To maintain sign consistency across bootstraps, if the first element of a bootstrapped $V^T$ was negative, that $V^T$ was multiplied by -1. Error regions depicted across figures (Fig. 3, S9) were the bounded volume of bootstrapped prime vectors (MATLAB *alphaShape* [Alpha parameter=10]).

Slope coefficients between pairs of measures was calculated as the ratio of corresponding values in $V^T$ (Fig. S4). Differences in slope coefficients between clusters was determined by subtracting bootstrapped slope coefficients for each cluster pair, yielding a distribution of 1000 bootstrapped slope differences per cluster and measure pair. After adjusting each distribution for narrowness bias (distributions extended from true slope differences by factor of $\sqrt{n/(n-1)}$)[31,32], the two-tailed p-value was determined as the proportion of percentile confidence intervals containing 0 (i.e., no difference between slopes).

*Stepwise logistic regression.* Multinomial logistic regression was used to assess whether retest cluster could be predicted from initial test clusters and/or self-report trait measures. Initial test cluster, CFI subscales (test, retest), HTQ subscales (test, retest), AUDIT (test, retest), and all 2-way interactions between variables were allowed to conditionally enter the model (likelihood ratio <0.05). Only initial cluster and intercept were included in the final model ($\chi^2_{(4)} = 32.82$, $p < 0.001$, Nagelkerke $r^2 = 0.262$).

**Post-retest reflections.** Cluster-related differences in test-retest awareness were assessed using one-way ANOVAs (Fig. S11). Between-subject factors were initial test cluster, retest cluster, or test-retest cluster change. Where applicable, follow-up pairwise between-subject comparisons (Sidak correction) were conducted.

## Reporting summary
Further information on research design is available in the Nature Portfolio Reporting Summary linked to this article.

## Results
### Distinct behavioral phenotypes of punishment sensitivity
Our international sample ($N = 267$; mean age = 32.17, $SD = 10.74$; $n = 118$ female, 143 male, 6 other; Fig. S1) participated in an online game ("Planets & Pirates") designed to measure punishment learning and decision-making[5]. During the task, participants could click on two continuously-available planets (R1, R2), with each response rewarded 50% of the time (Fig. 1a). After initial reward training, ship cues were introduced (punishment phase). Responses on R1 triggered a cue (R1 → CS+) predicting significant point loss ("attack"), while responses on R2 (R2 → CS) had no further consequences. Participants were randomly assigned to either a high-probability punishment group (40% CS probability; $n = 134$) or a low-probability group (10% CS probability; $n = 133$). To equalize the negative utility of R1 across groups, the 40% group experienced mild point loss from attacks (-10% of total points), and the 10% group faced severe loss (-40% of total points). R1 responses generally resulted in net point loss during the punishment blocks, so avoiding R1 was crucial for maximizing gains.

Following three punishment blocks, participants were given explicit corrective information regarding task contingencies (i.e., R1 → CS+ → Attack and R2 → CS–). They were also required to pass a contingency knowledge test to confirm they understood this information before completing a final game block (Rev) designed to assess how this information influenced their behavior.

Before punishment, participants showed no preference between R1 and R2 (Fig. 1b; see also Fig. S2) ($t_{(266)} = 0.058$, $p = 0.953$, $d = 0.004$ [95% CI:–0.12,12], $BF_{01} = 14.6$). However, as expected, avoidance of punished R1 increased across punishment blocks ($F_{(1,264)} = 402.19$, $p < 0.001$). The distribution of punishment avoidance was bimodal (Fig. 1b). Using automated clustering, we identified three distinct behavioral phenotypes: "Sensitive" ($n = 70$), who learned to avoid R1 before the contingency reveal (Pun3: $t_{(69)} = -51.53$, $p < .001$, $d = -6.16$ [95%CI:–7.2,–5.1]); "Unaware" ($n = 126$), who only avoided after the reveal (Pun3: $t_{(125)} = -1.19$, $p = 0.238$, $d = -0.106$ [95%CI:–2.8,0.07], $BF_{01} = 5.09$; Rev: $t_{(125)} = -124.1$, $p < 0.001$, $d = -11.06$ [95%CI:–12.4,–9.7]); and "Compulsive" ($n = 71$), who did not substantially avoid R1 even after explicit information was provided (Pun3: $t_{(57)} = -1.79$, $p = 0.078$, $d = -0.213$ [95%CI:–0.45,0.02], $BF_{01} = 1.69$; Rev: $t_{(57)} = -10.91$, $p < 0.001$, $d = -1.30$ [95%CI:–1.6,–0.98]). These phenotypes were observed across demographic variables, but Compulsives were over-represented older individuals ($\chi^2_{(4)} = 12.42$, $p = 0.014$) (Fig. S1b).

Phenotypes did not markedly differ in their overall click rates (cluster: $F_{(2264)} = 3.317$, $p = 0.038$, $\eta_p^2 = 0.025$ [90%CI: 0,0.059]; block*cluster: $F_{(2264)} = 2.201$, p = 0.113, $\eta_p^2 = 0.025$ [90%CI:0,0.045]) (Fig. S3a), indicating

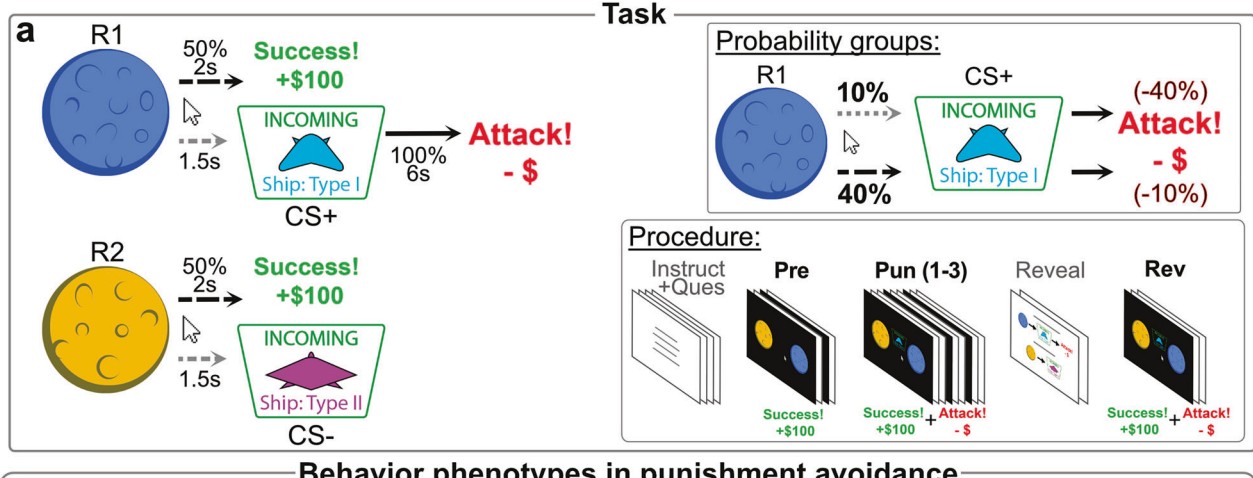

**a** Task

**b** Behavior phenotypes in punishment avoidance

**c**

**d**

similar engagement and motivation in the task, but they employed markedly different response strategies (action*cluster [pre-reveal]: $F_{(2,264)} = 26.82$, $p < 0.001$, $\eta_p^2 = 0.169$ [90%CI: 0.102,0.232]). Sensitive participants shifted their responses away from the punished R1 toward the unpunished R2 during punishment blocks (action*block: $F_{(1,69)} = 32.58$, $p < 0.001$, $\eta_p^2 = 0.321$ [90%CI: 0.174,0.443]), whereas Unawares (action*block:

$F_{(1,125)} < 0.001$, $p = 0.988$, $\eta_p^2 < 0.001$ [90%CI: 0,1]) and Compulsives (action*block: $F_{(1,70)} = 0.182$, $p = 0.671$, $\eta_p^2 = 0.003$ [90%CI: 0,0.053]) participants did not. This difference had a significant impact on outcomes (block*cluster: $F_{(2264)} = 78.618$, $p < 0.001$, $\eta_p^2 = 0.229$ [90%CI: 0.297,0.436]): Sensitive participants accumulated points during punishment blocks, while Unaware and Compulsive participants generally lost points (Figs. 1c, S3b).

**Fig. 1 | Behavioral phenotypes in punishment avoidance. a** Conditioned punishment task design. Across game blocks, participants could click on continuously-available planets (R1, R2) for point rewards. During punishment phase (*Pun [1–3]*), these responses also led to ship cues (CS+ vs. CS−) with 10% or 40% probability (assigned probability group). CS+ led to an "Attack" (-40% or -10% point loss, according to probability group), whereas CS− was safe. The R1 → CS+ → Attack punishment contingency meant R1 had negative utility (matched across probability groups); R1 needed to be avoided to maximize point gain. Contingencies (R1 → CS + → Attack; R2 → CS−) were revealed to participants before the final punished game block (*Rev*). **b** Mean (±SEM) [left panel] and individual [middle, right panels]

R1 preferences per behavioral phenotype across blocks. * $p < 0.05$ one sample $t$ test vs. 50% (no R1:R2 bias). **c** Mean (±SEM) point gain across blocks. Poor avoidance during punishment was associated with attenuated point gain (Pun 2: $F_{(2, 264)} = 5.098$, $p = 0.007$; Pun 3: $F_{(2,264)} = 56.243$, $p < 0.001$; Rev: $F_{(2,264)} = 76.91$, $p < 0.001$). **d** Composition of phenotypes by probability group [top panel], and vice versa [bottom panel] ($\chi^2_{(2)} = 22.61$, $p < 0.001$). Stronger contingencies drove individuals towards a Sensitive phenotype ($\chi^2_{(1)} = 16.24$, $p < 0.001$), whereas weaker contingencies drove individuals towards Compulsive ($\chi^2_{(1)} = 4.208$, $p = 0.040$) over Unaware ($\chi^2_{(1)} = 2.162$, $p = 0.142$) phenotype.

Therefore, despite continued effort, the strategies of Unaware and Compulsive phenotypes were counterproductive.

We examined the impact of punishment probability group on behavior. Consistent with expectations, mild but frequent punishment led to more avoidance than severe but infrequent punishment (group [pre-reveal]: $F_{(1265)} = 17.18$, $p < 0.001$, $\eta_p^2 = 0.061$ [90%CI: 0.022,0.112]) (Fig. S3c). Furthermore, punishment probability influenced behavioral phenotype distribution (Fig. 1d). Higher punishment probabilities increased the likelihood of being categorized as Sensitive, while lower probabilities increased the likelihood of being categorized as Compulsive ($\chi^2_{(2)} = 22.61$, $p < 0.001$). Importantly, punishment probability did not have a significant effect after accounting for phenotype (group [pre-reveal]: $F_{(1,261)} = 2.226$, $p = 0.137$, $\eta_p^2 = 0.008$ [90%CI: 0,0.036], BF$_{01} = 4.19$; group [Rev]: $F_{(1,261)} = 0.002$, $p = 0.967$, $\eta_p^2 < 0.001$ [90%CI: 0,1], BF$_{01} = 9.4$) (Fig. S3c), showing the effect of punishment probability on behavior is mediated by its influence on phenotype.

**Punishment phenotypes do not differ in habit formation, outcome valuations, or cue-related learning.** These findings show that a significant proportion of individuals fail to modify their behavior to avoid punishment, even after receiving repeated punishers and explicit information on how to do so. To identify the mechanisms underlying these individual differences in learning and decision-making, we first examined several commonly proposed explanations[19].

One possibility is that punishment insensitive behavior may arise from habit formation, where actions become automatic and disconnected from their consequences[33]. Two key characteristics of habits—automaticity and independence from goals—were assessed after each block. Participants estimated their R1:R2 preference for the previous block (preference estimate) and what R1:R2 preference they believed would have maximized their point gain (preference endorsement). Comparing these with actual behavior allowed us to assess participants' awareness of their own actions and whether their choices aligned with their task goals. Contrary to a habit-based explanation, preference estimates (Fig. 2b) and endorsements (Fig. 2c) closely matched actual behavioral preference (Fig. 2a) across all phenotypes (cluster*measure: BF$_{01} = 15.92$). Each phenotype showed accurate awareness of what they chose (estimates) and why they chose it (endorsements), stating their own choices as optimal for maximizing points, even when they were not. This shows that punishment-insensitive behavior was deliberate and self-assured (i.e., goal-directed), not habitual.

A second possibility is that punishment insensitive behavior may be driven by distortions in value-based decision-making, where individuals overvalue rewards or undervalue punishments, leading to persistent engagement in punished actions[34]. To test this, we assessed participants' valuations of rewards and punishments. All phenotypes reported similar levels of liking for rewards (cluster: $F_{(2264)} = 3.975$, $p = 0.02$, $\eta_p^2 = 0.029$ [90%CI: 0.003,0.066]) and disliking for punishments (cluster: $F_{(2264)} = 0.502$, $p = 0.606$, $\eta_p^2 = 0.004$ [90%CI: 0,0.019], BF$_{01} = 14.36$) (Fig. 2d), indicating that the differences in punishment avoidance were not due to distorted reward or punishment valuation in the conditions tested.

A third explanation is that individuals may fail to learn about environmental cues (e.g., the CS+ ship cue) that predict punishment and mediate the relationship between actions and point loss[35]. To investigate this, we measured how participants valued the ship cues (CSs) and their attribution

of attacks to these cues. All phenotypes quickly developed a dislike for CS+ over CS− (CS*block [pre-reveal]: $F_{(1,264)} = 214.38$, $p < 0.001$, $\eta_p^2 = 0.448$ [90%CI: 0.377,0.508]; CS*block*cluster [pre-reveal]: $F_{(2,264)} = 2.691$, $p = 0.070$, $\eta_p^2 = 0.02$ [90%CI: 0,0.051], BF$_{01} = 35.43$) (Fig. 2e), reflecting an understanding of the Pavlovian CS→Attack contingencies (Fig. 2g). This indicates that all participants readily learned about and appropriately valued the cues predicting point loss. Therefore, the pronounced differences in behavior between phenotypes cannot be attributed to failures in general learning or motivation.

**Punishment phenotypes reflect different cognitive-behavioral trajectories.** Differences in punishment sensitivity were instead driven by what individuals learned about their actions. Phenotypes exhibited stark differences in how they valued each action during the task (action*block*cluster [pre-reveal]: $F_{(2264)} = 102.61$, $p < 0.001$, $\eta_p^2 = 0.437$ [90%CI: 0.363,0.496]) (Fig. 2f). While actions were valued similarly before punishment (action*cluster: $F_{(2264)} = 1.872$, $p = 0.156$, $\eta_p^2 = 0.014$ [90%CI: 0,0.041], BF$_{01} = 4.59$), only the Sensitive group showed a clear preference for the unpunished action (R2) over the punished action (R1) during the punishment blocks (action [Sensitive]: $F_{(1,69)} = 203.2$, $p < 0.001$, $\eta_p^2 = 0.747$ [90%CI: 0.656,0.799]). In contrast, Unaware and Compulsive groups failed to discriminate ([Unaware]: $F_{(1,125)} = 2.585$, $p = 0.110$, $\eta_p^2 = 0.02$ [90%CI: 0,0.077], BF$_{01} = 0.927$; [Compulsive]: $F_{(1,70)} = 1.064$, $p = 0.306$, $\eta_p^2 = 0.015$ [90%CI: 0,0.091], BF$_{01} = 7.29$). This difference reflected each group's ability to form accurate Action→CS (correct*cluster [pre-reveal]: $F_{(2,264)} = 83.92$, $p < 0.001$, $\eta_p^2 = .389$ [90%CI: 0.312,0.451]) and Action→Attack inferences (action*cluster: $F_{(2,264)} = 38.92$, $p < 0.001$, $\eta_p^2 = 0.228$ [90%CI: 0.155,0.293]) (Fig. 2h, S3d). Sensitives correctly attributed CS+ and attacks to the punished action (correct [pre-reveal]: $F_{(1,69)} = 181.4$, $p < 0.001$, $\eta_p^2 = 0.724$ [90%CI: 0.627,0.781]; action: $F_{(1,69)} = 94.57$, $p < 0.001$, $\eta_p^2 = 0.578$ [90%CI: 0.446,0.663]), whereas Unawares (correct: $F_{(1,125)} = 4.612$, $p = 0.034$, $\eta_p^2 = 0.036$ [90%CI: 0.001,0.102]; action: $F_{(1,125)} = 3.371$, $p = 0.069$, $\eta_p^2 = 0.026$ [90%CI: 0,0.087], BF$_{01} = 1.68$) and Compulsives (correct: $F_{(1,70)} = 6.601$, $p = 0.012$, $\eta_p^2 = 0.086$ [90%CI: 0.010,0.200]; action: $F_{(1,70)} = 0.371$, $p = 0.544$, $\eta_p^2 = 0.005$ [90%CI: 0,0.065], BF$_{01} = 3.96$) largely did not. Consequently, Sensitives learned to value the unpunished action, while the other groups did not.

Providing explicit, correct contingency information corrected these knowledge deficits in Unawares, leading to a pronounced revaluation of actions (Fig. 2f) and improved decision-making (Fig. 1b). Although Compulsives showed some cognitive updating (Fig. 2f), it was significantly attenuated (action*block*cluster [Unaware vs. Compulsive]: $F_{(1,195)} = 53.71$, $p < 0.001$, $\eta_p^2 = 0.216$ [90%CI: 0.136,0.295]); they continued to incorrectly attribute punishment to the unpunished action, despite receiving the same corrective information and passing the same assessment of punishment contingency knowledge.

To further understand these differences, we mapped the interrelationships between action-related cognitions and behavior (i.e., R1:R2 punishment inferences, valuations, endorsements, estimates and behavior) across blocks for each phenotype using singular value decomposition[36] (Fig. 3). This analysis reveals the latent cognitive-behavioral trajectory of each phenotype to show whether and how phenotypes differ in their cognitive-behavioral integration. We hypothesized that punishment inferences influence action values, which in turn affect endorsed behavioral

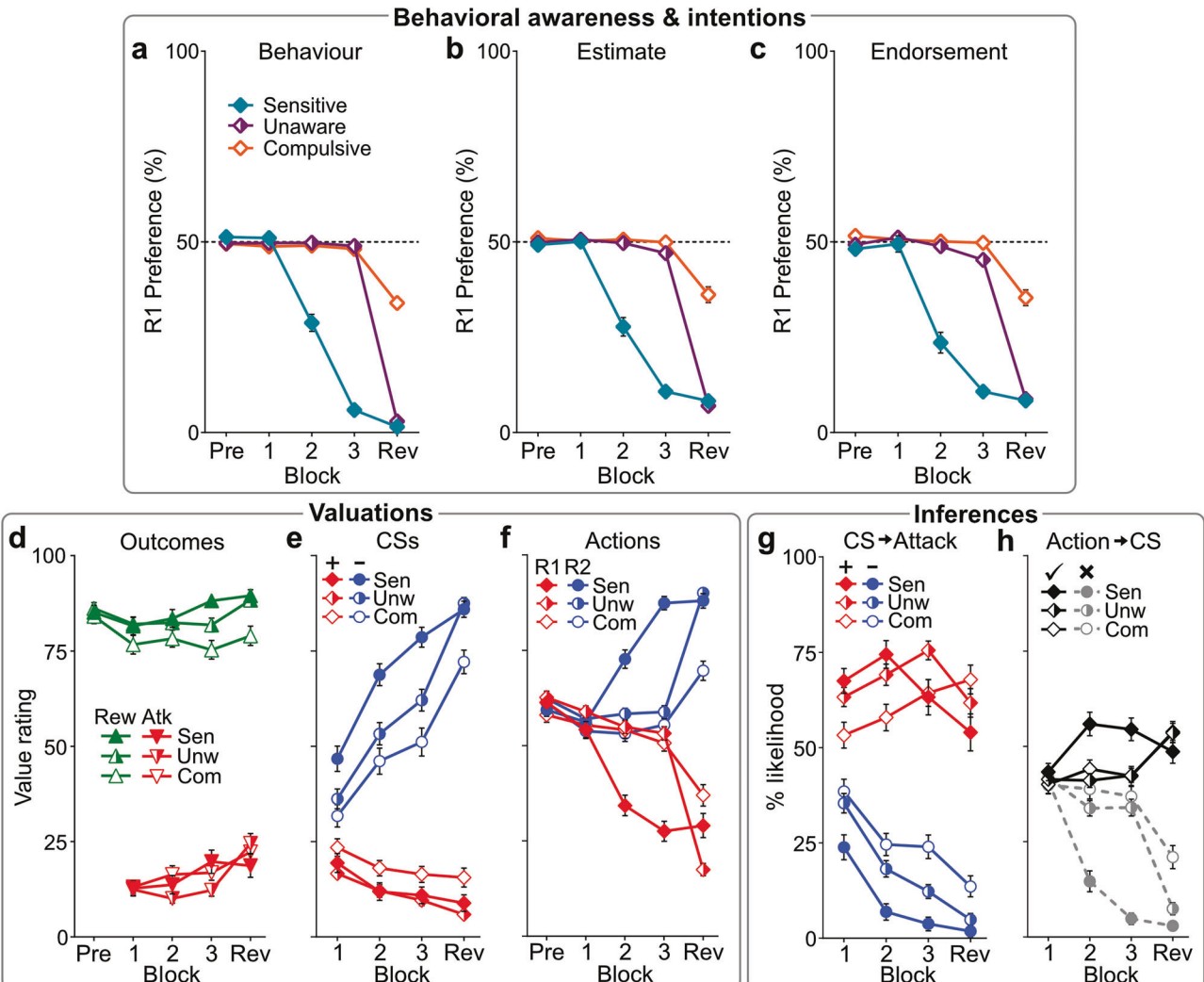

**Fig. 2 | Psychological underpinnings of behavioral phenotypes.** Mean (±SEM) R1:R2 behavioral preference (**a**), post-block estimates of R1:R2 preference (**b**), and post-block endorsed (perceived optimal) R1:R2 preference (**c**) per phenotype across blocks. Behavior estimates and endorsements matched actual behavior, suggesting behavior across phenotypes was deliberate, not habit-like. **d–f** Mean (±SEM) value ratings for point outcomes (reward [Rew], attack [Atk]), CSs (CS+, CS−), and actions (R1, R2) across blocks. **d** All phenotypes reported liking rewards (cluster: $F_{(2,264)} = 3.975$, $p = 0.02$) and disliking attacks (cluster: $F_{(2,264)} = 0.502$, $p = 0.606$). **e** All phenotypes rapidly disliked CS+ relative to CS− before contingency reveal (CS*block [pre-reveal]: $F_{(1,264)} = 214.38$, $p < 0.001$; CS*block*cluster [pre-reveal]: $F_{(2,264)} = 2.691$, $p = 0.070$). **f** Only Sensitives valued actions differently prior to contingency reveal (action*block*cluster [pre-reveal]: $F_{(2,264)} = 102.61$, $p < 0.001$; action [Sensitive]: $F_{(1,69)} = 203.2$, $p < 0.001$; [Unaware]: $F_{(1,125)} = 2.585$, $p = 0.110$; [Compulsive]: $F_{(1,70)} = 1.064$, $p = 0.306$). The reveal drove stronger action

revaluation for Unawares than Compulsives (action*block*cluster [Unaware vs. Compulsive]: $F_{(1,195)} = 53.709$, $p < 0.001$); Compulsives continued to undervalue unpunished actions after the reveal ($p < 0.001$ vs. Unawares and Sensitives). **g, h** Mean (±SEM) causal inferences per phenotype across blocks. **g** All clusters were broadly aware that CS+, not CS−, led to Attack before the reveal (CS*block [pre-reveal]: $F_{(1,264)} = 60.53$, $p < 0.001$; CS*block*cluster: $F_{(2,264)} = 3.282$, $p = 0.039$). **h** Only Sensitives developed accurate Action→CS (correct ✓ [R1 → CS+ ; R2 → CS−] over incorrect ✗ [R1 → CS−; R2 → CS+]) inferences before the reveal (correct*block [Sensitives]: $F_{(1,69)} = 147.0$, $p < .001$; [Unaware]: $F_{(1,125)} = 3.380$, $p = 0.068$; [Compulsive]: $F_{(1,70)} = 2.748$, $p = 0.102$). Contingency reveal drove stronger inference updating for Unawares than Compulsives (correct*block*cluster [Unaware vs. Compulsive]: $F_{(1,195)} = 4.816$, $p = 0.029$); Compulsives continued to misattribute attacks to the unpunished action after the reveal ([vs. Unawares, Sensitives]: $p < 0.001$).

strategies (i.e., intentions). Assuming intact behavioral control and awareness, these intentions should dictate actual behavior and post hoc estimates of behavior (Fig. 3f).

Interestingly, Sensitives and Unawares shared similar trajectories (Fig. 3a–e), despite their initial differences. Once Unawares received contingency information, they used this to correct their punishment-related knowledge, which resulted in corresponding changes to action valuations, preference endorsements, behavior, and estimates (Figs. 3f, S5). So, information-driven updating in Unawares aligned with experience-driven updating in Sensitives.

In contrast, Compulsives followed a distinct cognitive-behavioral trajectory. Unlike the other phenotypes, Compulsives exhibited a

divergence between their punishment inferences, action values, and behavior (Figs. 3a, b, S4a, S5a). Even though Compulsives used contingency information to modestly update their punishment inferences and valuations, these changes did not result in corresponding changes to endorsed or implemented behavior (Figs. 3a, b, S4a). So, attenuated punishment avoidance in Compulsives was not simply due to failures in belief updating. Rather, Compulsives were on an altered cognitive-behavioral trajectory, such that information-driven changes to action-related inferences and values did not readily translate to a corresponding behavioral strategy. This represents a perturbed integration of cognitive and behavioral processes, specifically at the junction between beliefs and intentions (Fig. 3f).

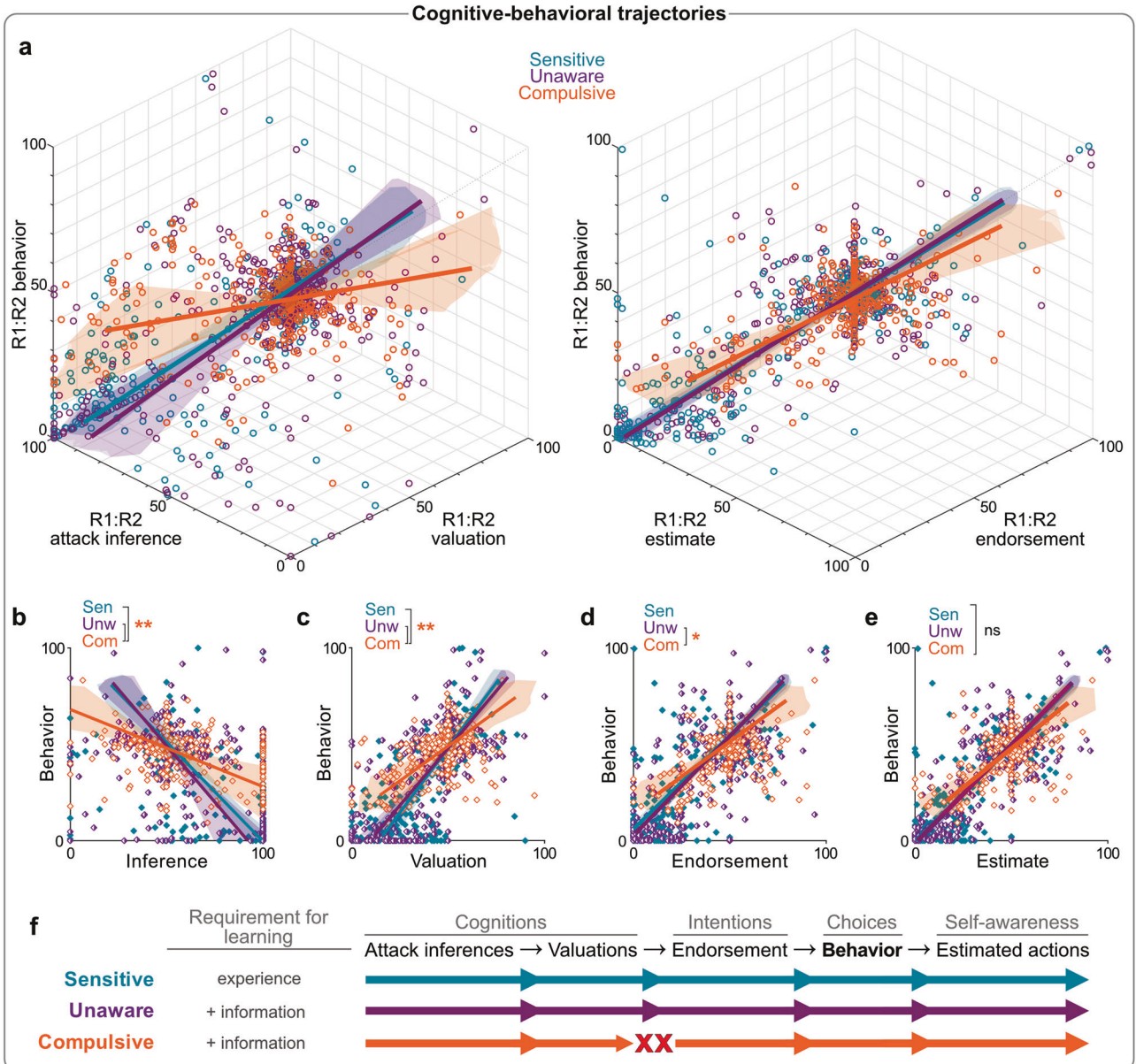

**Fig. 3 | Cognitive-behavioral trajectories of behavioral phenotypes. a** True-score relationships between Action-Attack inference bias, action value bias, and behavior preference [left panel], and endorsed preference, estimated preference, and behavior preference [right panel]. Colored lines represent true-score relationships per phenotype (determined via singular value decomposition), shaded regions represent 3D confidence regions for true score relationships (determined via bootstrapping), and each dot represents an individual's block score. Corresponding pairwise true-score relationships between behavior preference and Action-Attack inference bias (**b**), action value bias (**c**), endorsed preference (**d**), and estimated preference (**e**). R1:R2 attack inferences and R1:R2 valuations failed to translate to behavioral preference in Compulsives relative to other phenotypes, whereas relationships between endorsements, estimates and behavior were relatively intact. *$p < 0.05$; **$p < 0.01$ for cluster slope differences. **f** Model for underlying differences in behavioral phenotypes. Sensitive and Unaware individuals share cognitive-behavioral trajectories (functional relationship between cognitions, intentions, and behavior), but only Sensitives readily learn from experience alone. Compulsive individuals have an altered cognitive-behavioral trajectory: provision of information leads to updated cognitions (action-related inferences and valuations), but these do not translate into a corresponding endorsed behavioral strategy. $N = 267$ participants.

**Punishment phenotypes are trait-like.** To assess the stability of the punishment phenotypes, we retested participants after a 6-month interval ($N = 128$). No selective attrition was observed across original probability groups ($\chi^2_{(1)} = 0.63$, $p = 0.427$) and phenotypes ($\chi^2_{(2)} = 0.705$, $p = 0.703$) (Figure S6). Participants were reassigned to their original probability group but were randomly allocated to counterbalancing conditions (i.e., unlikely to have the same punished action, CS+ cue, etc).

Clustering of avoidance behavior at retest revealed the same Sensitive, Unaware, and Compulsive phenotypes as found in initial test (Fig. 4a). Additionally, the cognitive and motivational characteristics associated with each phenotype (e.g., poor Action-CS awareness, distinct cognitive-

behavioral trajectories) were recapitulated at retest (Figs. S4, S7–S9). These findings show the robustness of these phenotypes. Crucially, each phenotype at retest predominantly consisted of participants who exhibited the same phenotype at the initial test ($\chi^2_{(4)} = 30.811$, $p < 0.001$) (Fig. 4b, c), indicating that inter-individual differences are relatively stable over time. This suggests that variation in punishment learning and cognitive-behavioral integration reflect stable, trait-like characteristics.

Phenotype stability was related to self-reported cognitive flexibility ($F_{(2,125)} = 4.278$, $p = 0.016$, $\eta_p^2 = 0.064$ [90%CI: 0.007,0.134]). Participants who performed worse at retest (e.g., classified as Sensitive at initial test but Unaware at retest) reported significantly lower cognitive flexibility

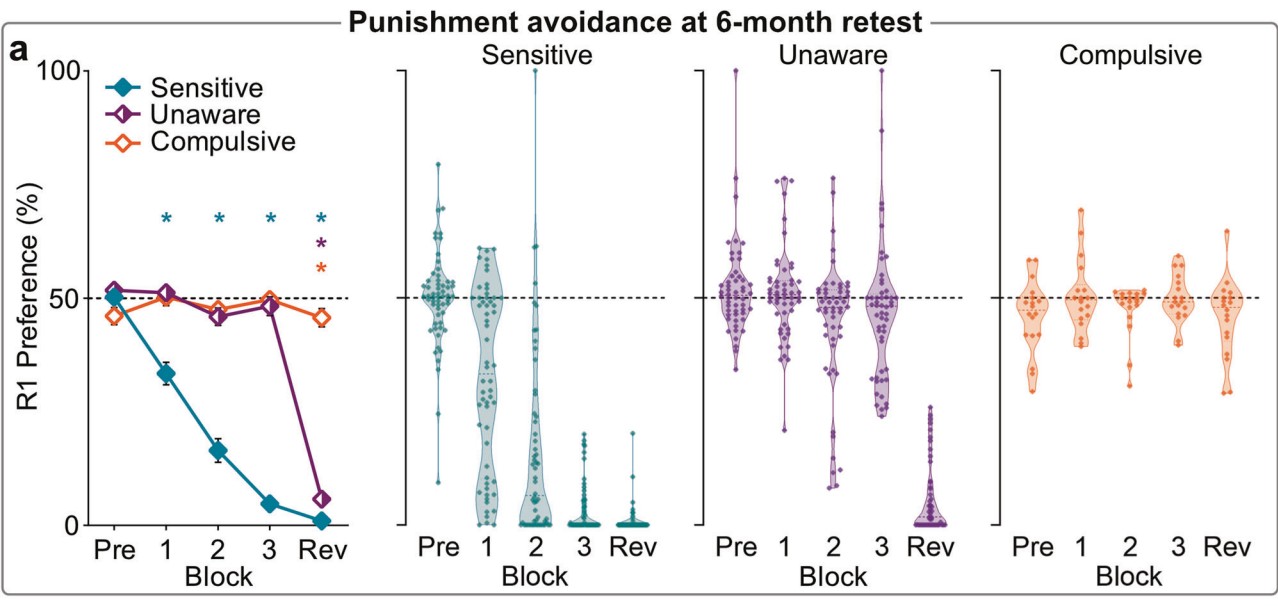

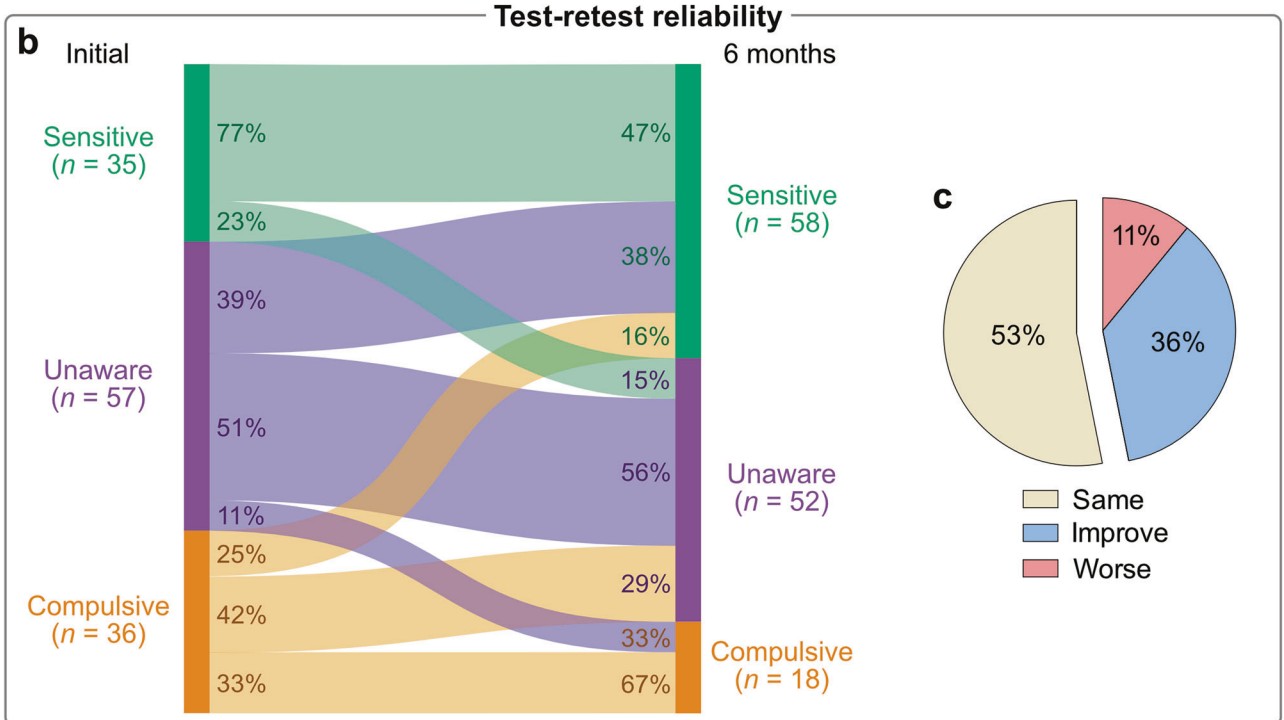

**Fig. 4 | Behavioral phenotypes at 6-month retest. a** Mean (±SEM) [left panel] and individual [middle, right panels] R1 preferences across blocks per retest cluster ($N = 128$). *$p < 0.05$ one sample $t$ test vs. 50% (no R1:R2 bias). **b** Relationship between phenotype at initial test and phenotype at retest. Individuals were significantly more likely to be categorized into the same phenotype at retest ($\chi^2_{(4)} = 30.811$, $p < 0.001$). In a stepwise logistic regression model, original phenotype

was the dominant predictor of retest phenotype ($\chi^2_{(4)} = 32.82$, $p < 0.001$, Nagelkerke $r^2 = 0.262$). **c** The majority of individuals had the same phenotype at retest. Some individuals improved their punishment phenotype (e.g., went from Unaware at initial test to Sensitive at retest), while some individuals were categorized as being in a worse punishment-avoiding cluster.

compared to those who maintained their performance ($MD = -11.67$, $p = 0.017$). However, cognitive flexibility was not significantly related to maintained versus improved performance across tests ($MD = 4.13$, $p = 0.341$), and phenotype stability was not significantly related to other self-reported tendencies previously linked to poor decision-making (Habitual Tendencies Questionnaire [$F_{(2,125)} = 0.55$, $p = 0.579$, $\eta_p^2 = 0.009$ [90%CI: 0,0.042]; Alcohol Use Disorders Identification Test [$F_{(2,125)} = 0.037$, $p = 0.946$, $\eta_p^2 = 0.001$ [90%CI: 0,1]) (Fig. S10).

Finally, we examined what combination of trait variables best predicted behavioral phenotype at retest. Self-reported traits across tests, initial test

phenotype (i.e., behaviorally-identified trait), and interactions between these, were submitted to a stepwise logistic regression model. Interestingly, initial phenotypes, but not self-report measures or interaction terms, were included in the final model (see Methods). So, behaviorally-identified phenotypes surpass traditional self-report measures in predicting future punishment learning and decision-making.

## Discussion
In a moderately large, diverse sample, we identified three enduring phenotypes of punishment sensitivity: Sensitives who adaptively avoid

punishment through experience, Unawares who require explicit information to correct maladaptive choices, and Compulsives who persist in punished behavior despite both experiential and informational interventions. These phenotypes reflect stable, trait-like differences and predicted longitudinal decision-making outcomes, positioning them as robust markers for persistent harmful behavior.

We identify two dissociable cognitive mechanisms driving these phenotypes. First, causal inference deficits drive punishment insensitivity and maladaptive behavior in Unawares. These individuals misattributed punishment, failing to correctly link their specific action to the delivery of punishment. These causal inference deficits explain why experiential feedback often fails to drive behavior change[37] - a finding that has proved challenging to policies relying on punishments (e.g., fines, sanctions, etc) to correct behavior[38–40]. Importantly, these causal inference deficits were remediable via an information-based intervention. Second, failures in cognitive-behavioral integration in Compulsives rendered punishment inert. Even with accurate causal knowledge (achieved post-intervention), these individuals could not synthesize their causal knowledge with action selection processes to avoid further punishment. This integration failure, driving resistance to both punishment experience and remedial information, identifies a core mechanism explaining the limitation of fact-based interventions (e.g., warnings, public health campaigns[37,41,42]) at correcting behavior in some individuals and suggests that suboptimal choices in these individuals may require alternative approaches.

These cognitive mechanisms exhibited striking stability within an individual. Behavioral phenotyping at baseline predicted individuals' learning trajectories and choice patterns six months later. Moreover, behavioral phenotyping outperformed self-reported cognitive and behavioral flexibility in forecasting long-term learning and decision-making outcomes, underscoring its potential as an objective marker for assessing the cognitive mechanisms that place individuals at risk of chronic, maladaptive decision-making. By mapping stable punishment decision-making phenotypes to distinct cognitive mechanisms, our work advances a framework to stratify punishment sensitivity (e.g., distinguishing misattribution from integration deficits)[19] and prioritize mechanism-targeted interventions that address the different cognitive drivers of poor decision-making[43].

## Limitations

While our findings provide insight into the cognitive mechanisms underpinning punishment sensitivity, several limitations warrant acknowledgment. First, although we identify distinct behavioral phenotypes that are stable over time, the task was conducted in a controlled online environment. As such, the ecological validity of these findings, including their relationship to more complex real-world decision-making, remains undetermined. Second, while our international sample was demographically diverse and drawn from 24 countries, data collection relied on an online participant pool fluent in English with access to digital devices. This may limit generalizability to populations with distinct socioeconomic and cultural profiles. Future work could examine whether these phenotypes also occur in more heterogeneous samples.

## Data availability

All experiment data is available at https://osf.io/ju35h/. Anonymized data have been deposited at https://osf.io/ju35h/. All experiment code and analysis scripts are accessible at https://github.com/philjrdb/HCP-Test-Retest and https://zenodo.org/records/155819599.

## Code availability

Code for data extraction and pre-processing can be found at https://github.com/philjrdb/HCP-Test-Retest and https://zenodo.org/records/155819599.

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

## Acknowledgements
This work was supported by grants from the Australian Research Council to PJRDB (DP220102317), G.P.M. (DP250100345; DP220100040), and by an NHMRC Synergy grant (GNT2011633). The funders had no role in study design, data collection and analysis, decision to publish or preparation of the manuscript.

## Author contributions
L.Z.: Conceptualization, Methodology, Investigation, Writing—review & editing. H.P.: Methodology, Visualization, Writing—review & editing. G.P.M.: Conceptualization, Methodology, Investigation, Funding acquisition, Project administration, Supervision, Writing—original draft, Writing—review & editing. P.J.R.D.B.: Conceptualization, Methodology, Investigation, Funding acquisition, Project administration, Supervision, Writing—original draft, Writing—review & editing.

## Competing interests
The authors declare no competing interests.
