## [Transparent Peer Review file · Communications Psychology]

Causal inference and cognitive-behavioral integration deficits drive stable variation in human punishment sensitivity

Corresponding Author: Dr Philip Jean-Richard-Dit-Bressel

Version 0:

Decision Letter:

Dear Professor McNally,

Thank you for your patience during the peer-review process. Your manuscript titled "Causal inference deficits and integration failure drive stable variation in human punishment sensitivity" has now been seen by 2 reviewers, and I include their comments at the end of this message. They find your work of interest but raised some important points. We are interested in the possibility of publishing your study in Communications Psychology, but would like to consider your responses to these concerns and assess a revised manuscript before we make a final decision on publication.

We therefore invite you to revise and resubmit your manuscript, along with a point-by-point response to the reviewers. Please highlight all changes in the manuscript text file.

Editorially, we consider that one aspect needs to be carefully addressed: the alternative interpretations of the "compulsive" group. This aspect comes across clearly in the assessment of both reviewers, and both suggest a set of supplementary analyses that could be done to address potential confounds.

While revising the manuscript please ensure you follow our statistical guidelines when reporting statistics (<https://www.nature.com/commpsychol/submit/submission-guidelines#statistical-guidelines>). Please note in particular our requirements for the reporting and interpretation of null-results. Non-significant findings derived from null-hypotheses significance tests should be reported in full, but may not be interpreted. Where you interpret null results, this interpretation must be based on Bayes Factors or equivalence tests.

I am attaching an Editorial Requests Table that details critical reporting requirements for the revised manuscript. Please attend to each item and ensure your manuscript is fully compliant. If your revised manuscript is not aligned with these requests on major issues, such as those concerning statistics, it may be returned to you for further revisions without re-review.

Please submit the following items:

- Revised manuscript
- Point-by-point response to the referees' comments
- Cover letter (as a separate document)
- <https://www.nature.com/documents/nr-reporting-summary.zip>>Nature Research Reporting Summary
- <https://www.nature.com/documents/nr-editorial-policy-checklist.pdf>>Editorial Policy Checklist

- Completed Editorial Request Table (attached).

via this link: Link Redacted .

Additional guidance is available in our style and formatting guide Communications Psychology formatting guide.

Best regards,

Eva R. Pool

Eva R. Pool, PhD
Editorial Board Member
Communications Psychology
orcid.org/0000-0001-5929-1007

REVIEWER EXPERTISE:

Reviewer #1 Computational model of cognition, individual differences
Reviewer #2 Computational model of cognition, psychiatry

REVIEWER REPORTS:

Reviewer #1 (Remarks to the Author):

This study aims to characterize behavioral phenotypes that emerge during punishment learning. Using a task where participants first learn reward contingencies before having these devalued via pairing with a punishment, the authors demonstrate that three phenotypes emerge: one group learns to avoid the punished stimulus, one learns this only after explicit instruction, and a third never learns to avoid the punished stimulus.

The work is thorough and tests various explanations for the results to show that this pattern is underpinned by differences in awareness of the action > CS- contingency. Notably, these phenotypes also appear to be largely stable over 6 months. On the whole, this is a novel, robust, and interesting study that addresses an important question regarding perseverance in the face of punishment.

I have only a few suggestions for improvement below.

1. I think it would be helpful to include some more statistics in the text, as it is sometimes hard to follow how exactly various results were determined (e.g., "Consistent with expectations, severe but infrequent punishment led to less avoidance than mild but frequent punishment")
2. I'm curious whether the inability to learn about the action > CS- contingency in the compulsive group reflects a general learning impairment. Were there any differences across the groups in learning of the original action > reward contingencies?
3. Another explanation could be that this is simply an effect of motivation, where the compulsive group simply have low motivation to complete the task and therefore do not learn the action > CS- contingency. Is there any evidence that would counter this?
4. Alternatively, could the high degree of "compulsive" behavior emerge as a result of the peculiarities of the task design? As both R1 and R2 are initially paired with 50% probability of reward, it is unnecessary to actually encode their identity (if I get reward 50% of the time regardless of where I click, there is no need to pay attention to their color). If this is the case, this could then impair learning on the punishment phase, since action > CS- learning will be more difficult if the participant is less

aware of the action identity.

5. Were demographic variables predictive of phenotype?

Reviewer #2 (Remarks to the Author):

This is an interesting study on individual differences in a simple instrumental learning task. Using a gamified task, an online setting, and a medium-sized sample, the authors demonstrate 3 distinct groups of participants with different learning trajectories. Cognitive underpinnings of these differences are followed up by meta-cognitive ratings, and the stability is followed in a 6-month retest.

While the paradigm is relatively simple, the categorical differences between individuals are impressive and bear real-life importance. As such, I'm convinced this presents and advancement of the field. The study is novel, original, and conducted with rigour. I would suggest the following improvements.

1. I missed a statistical (NHST or Bayesian) test of the plausibility of the three-group solution.
2. The sample is certainly not „large“ for an online-sample. Many lab-based studies routinely use N = 100 per experiment. This should be toned down and discussed.
3. Even though the authors took efforts to ensure a motivated sample (e.g. using Prolific and not Mturk), it is still possible that the „compulsive“ group is either inattentive (e.g. performing a second task on the side, as many online subjects do), and/or being more noisy in their behaviour. The issue is exhaustively discussed in Zorowitz et al. (2023) Nature Human Behaviour, who suggest specific analyses to account for this possibility. Such supporting analyses should be added and the matter discussed.
4. It would be important to mention that the study was not pre-registered.

Communications Psychology is committed to improving transparency in authorship. As part of our efforts in this direction, we are now requesting that all authors identified as 'corresponding author' create and link their Open Researcher and Contributor Identifier (ORCID) with their account on the Manuscript Tracking System prior to acceptance. ORCID helps the scientific community achieve unambiguous attribution of all scholarly contributions. You can create and link your ORCID from the home page of the Manuscript Tracking System by clicking on 'Modify my Springer Nature account' and following the instructions in the link below. Please also inform all co-authors that they can add their ORCID to their accounts and that they must do so prior to acceptance.

Version 1:

Decision Letter:

Dear Dr Jean-Richard-Dit-Bressel,

Your manuscript titled "Causal inference and cognitive-behavioral integration deficits drive stable variation in human punishment sensitivity" has now been seen by our reviewers, whose comments appear below. In light of their advice I am delighted to say that we are happy, in principle, to publish a suitably revised version in Communications Psychology.

We therefore invite you to revise your paper one last time to address the remaining concerns of our reviewers and a list of editorial requests. At the same time we ask that you edit your manuscript to comply with our format requirements and to maximise the accessibility and therefore the impact of your work.

EDITORIAL REQUESTS:

SUBMISSION INFORMATION:

OPEN ACCESS:

* DATA AVAILABILITY:

Link Redacted

Best regards,

Troy Lui

Troy Lui, PhD
Associate Editor
Communications Psychology

Eva R. Pool, PhD
Editorial Board Member
Communications Psychology
orcid.org/0000-0001-5929-1007

REVIEWERS' COMMENTS:

Reviewer #1 (Remarks to the Author):

The authors have addressed the comments I made very well with some additional analyses that satisfy any concerns I may have had around the conclusions. This is an interesting paper and I look forward to seeing it published.

Reviewer #2 (Remarks to the Author):

Thank you for giving me the opportunity to review this revision. The authors have addressed all my concerns.

Response to Reviewers

We thank both reviewers for their time and helpful comments. All changes to the manuscript are in **blue**. We respond to reviewers' comments in turn below.

Reviewer #1 (Remarks to the Author):

1. I think it would be helpful to include some more statistics in the text, as it is sometimes hard to follow how exactly various results were determined (e.g., “Consistent with expectations, severe but infrequent punishment led to less avoidance than mild but frequent punishment”)

We acknowledge that the lack of in-text citations was at times unhelpful. However, given the large volume of analyses included in this study, we opted to report all statistics in the figure legends. We believe this choice substantially improves the readability of the manuscript, and aligns with the statistical reporting guidelines provided by the journal.

Using the example sentence provided, the original text forwarded readers to Figure S1c (“...punishment led to less avoidance than mild but frequent punishment (**Figure S1c**)”) (now **Figure S3c**), which has the following in the figure legend:

“Low probability (but high severity) punishment caused less punishment avoidance than high probability (low severity) punishment (group [pre-reveal]: $F_{(1,265)}=17.18, p<.001, \eta_p^2=.061$).”

We agree that the original text lacked clarity, so we have amended it.

For sake of readability and consistency, we continue to report statistical analyses in the figure legends. This allows statistics to be reported in full and in a consistent location without it dominating the main text.

2. I’m curious whether the inability to learn about the action > CS- contingency in the compulsive group reflects a general learning impairment. Were there any differences across the groups in learning of the original action > reward contingencies?

There were no apparent learning deficits in the compulsive cluster prior to punishment phase. Compulsives had equivalent response rates, point gain, valuations, endorsements, and behaviour estimates (Figure 1b-c, 2, 4, **S2, S3, S7, S8**), and were located in the same multidimensional space (Figure **S5a, S9b**), during Pre-punishment blocks as other clusters.

We also report Action-Reward inferences across pre-punishment blocks in **Figure S2b**. No differences across clusters are observed.

3. Another explanation could be that this is simply an effect of motivation, where the compulsive group simply have low motivation to complete the task and therefore do not learn the action > CS- contingency. Is there any evidence that would counter this?

We agree that this is an important consideration.

Across the study, Compulsives had equivalently high response rates (planet-clicking for points) as other clusters (see Figure **S3, S4**), amounting to ~40 clicks/min. Furthermore, these participants maintained responding on both planets, which is arguably more effortful than only clicking on one planet (even when clicking on a single planet was beneficial). Compulsives were also largely able to report accurate CS+/CS-→Attack inferences, valuations of outcomes and CSs (Figure 2, **S8**), and accurately estimated their own behaviour across the task.

This, in addition to our engagement checks, indicate Compulsive individuals were sufficiently motivated. We note this in the main text (line 138-139: “Therefore, the pronounced differences in behavior between phenotypes cannot be attributed to failures in general learning or motivation.”). Nonetheless, we have added additional text to the manuscript to further highlight this.

4. Alternatively, could the high degree of “compulsive” behavior emerge as a result of the peculiarities of the task design? As both R1 and R2 are initially paired with 50% probability of reward, it is unnecessary to actual encode their identity (if I get reward 50% of the time regardless of where I click, there is no need to pay attention to their color). If this is the case,

this could then impair learning on the punishment phase, since action > CS- learning will be more difficult if the participant is less aware of the action identity.

We thank the reviewer for raising this interesting possibility. To address this, we have included additional analyses of behaviour, valuations, and reward inferences for the two pre-punishment blocks (new Figure S2).

Specifically, we show that even though there was no overall preference for R1 over R2 across pre-punishment blocks (as expected), there were random differences in how individuals responded on R1 vs R2 across these blocks. These spurious differences were related to individual perceptions of relative reward probability and action values (shown via Principal Component Analysis). That is, some individuals responded more on one planet than the other due to spurious perceptions of it being probabilistically more rewarding/valuable than the other. Corresponding to their spurious nature, these preferences were unrelated across blocks (i.e., a participant could infer greater reward from one planet in the 1st pre-punishment block, but greater reward from the other planet in the 2nd pre-punishment block). This pattern was observed for the compulsive cluster, indicating this cluster was distinguishing between the two responses and forming consistent discriminated beliefs regarding each action.

5. Were demographic variables predictive of phenotype?

We thank the reviewer for this important question. We now report demographic variables and their relationship to phenotypes in a new Figure S1.

In short, we found that phenotype changed across age brackets; older individuals were more likely to be compulsive (although all phenotypes were represented across age brackets).

No significant differences were found when examining relationships between gender and phenotype.

Reviewer #2 (Remarks to the Author):

1. I missed a statistical (NHST or Bayesian) test of the plausibility of the three-group solution.

We now report the Bayesian Information Criterion output for TwoStep clustering and relevant silhouette values in Table S1. The 3-cluster solution has the highest average silhouette value, indicating its plausibility.

2. The sample is certainly not „large“ for an online-sample. Many lab-based studies routinely use N = 100 per experiment. This should be toned down and discussed.

We agree this is overstated. We now refer to our sample ($N = 267$ after exclusions) as “moderately large”.

3. Even though the authors took efforts to ensure a motivated sample (e.g. using Prolific and not Mturk), it is still possible that the „compulsive“ group is either inattentive (e.g. performing a second ask on the side, as many online subjects do), and/or being more noisy in their behaviour. The issue is exhaustively discussed in Zorowitz et al. (2023) Nature Human Behaviour, who suggest specific analyses to account for this possibility. Such supporting analyses should be added and the matter discussed.

We agree the issue of inattentive responding is an important factor to consider. Our study took similar and additional steps to those outlined in Zorowitz et al (2023) to identify and remove inattentive responders from our sample. This is described in Methods (Participants subsection, line 415 onwards):

Participants were excluded from analyses if they failed either of two engagement checks: 1) failing to give correct responses to two catch questions embedded in the questionnaire battery (see *Self-reported trait questionnaires* for details); or 2) answering post-block measures too quickly or slowly (<0.8s or >30s per question, averaged per page).

We believe our approach of embedding the catch question within specific questionnaires (and deliberately matching catch question length to surrounding questionnaire items) substantially improves their ability to catch inattentive responders. The additional requirement of having nominal response times on post-block measures serves to catch those that may be attending to non-experiment tasks throughout the experiment.

We now reference Zorowitz et al (2023) to emphasize the importance of this issue and provide context for the steps taken to remove inattentive responders.

4. It would be important to mention that the study was not pre-registered.

We thank the reviewer for this suggestion, and have added text to Methods indicating this.